# The Ankyrin Repeat and Kinase Domain Containing 1 Gene Polymorphism (*ANKK1*
*Taq1A*) and Personality Traits in Addicted Subjects

**DOI:** 10.3390/ijerph16152687

**Published:** 2019-07-27

**Authors:** Anna Grzywacz, Jolanta Chmielowiec, Krzysztof Chmielowiec, Bożena Mroczek, Jolanta Masiak, Aleksandra Suchanecka, Olimpia Sipak-Szmigiel, Kamila Szumilas, Grzegorz Trybek

**Affiliations:** 1Independent Laboratory of Health Promotion of the Pomeranian Medical University in Szczecin, 11 Chlapowskiego St., 70-204 Szczecin, Poland; 2Department of Hygiene and Epidemiology, Faculty of Medicine and Health Science, University of Zielona Góra, 28 Zyty St., 65-046 Zielona Góra, Poland; 3Department of Human Sciences in Medicine, Pomeranian Medical University in Szczecin, 11 Chlapowskiego St., 70-204 Szczecin, Poland; 4Neurophysiological Independent Unit, Department of Psychiatry, Medical University of Lublin, 1 Aleje Racławickie St., 20-059 Lublin, Poland; 5Department of Obstetrics and Pathology of Pregnancy, Pomeranian Medical University in Szczecin, 48 Żołnierska St., 71-210 Szczecin, Poland; 6Department of General Pathology, Pomeranian Medical University, Szczecin 70-111, Poland; 7Department of Oral Surgery, Pomeranian Medical University in Szczecin, 72 Powstańców Wlkp. St., 70-111 Szczecin, Poland

**Keywords:** addiction, polymorphism, personality traits, polysubstance use disorder

## Abstract

The *Taq1A* polymorphism located in the *ANKK1* gene is one of the most widely studied polymorphisms in regards to the genetics of behavior and addiction. The aim of our study was to analyze this polymorphism with regard to personality characteristics and anxiety measured by means of the Personality Inventory—(NEO Five-Factor Inventory—NEO—FFI) and the State-Trait Anxiety Inventory (STAI) in polysubstance addicted subjects. The study group consisted of 600 male volunteers, including 299 addicted subjects and 301 controls. Psychiatrists recruited members for both groups. Addiction was diagnosed in the case group. In the control group mental illness was excluded. The same psychometric test and genotyping using the real-time PCR (polymerase chain reaction) method was performed for both groups. The results were investigated by means of multivariate analysis of the main effects Multi-factor ANOVA. Significantly higher scores on the scale of STAI state and Neuroticism and Openness traits, as well as lower scores on the scales of Extraversion, Agreeability, and Conscientiousness, were found in the case group subjects, compared to the controls. Differences in frequency of genotypes and alleles of *Taq1A* polymorphism between the studied groups were not found. Multi-factor ANOVA of addicted subjects and control subjects and the ANKK1 Taq1A variant interaction approximated the statistical significance for the STAI state. The main effects ANOVA of both subjects’ groups were found for the STAI state and trait, the Neuroticism scale, the Extraversion scale, and the Agreeability scale. The *ANKK1 Taq1A* main effects approximated the statistical significance of the STAI trait. Our study shows not only differences in personality traits between addicted and non-addicted subjects, but also the possible impact of *ANKK1* on given traits and on addiction itself.

## 1. Introduction

One of the most widely studied genetic variants associated with addictions and other psychiatric disorders is *Taq1A,* a single nucleotide polymorphism (SNP, rs1800497) [1,2,3]. *Taq1A* is located in the coding region of the ankyrin repeat and kinase domain, including 1 (*ANKK1*) gene that is adjacent to the *DRD2* gene coding the dopamine D2 receptor (D2R) [4]. This polymorphism results in the conversion of glutamine to lysine at locus 731 of the encoded protein. The *ANKK1* gene contains 8 exons and encodes a protein with 765 amino acids which are involved in the processes of data transformation in the central nervous system (CNS) [4,5]. There are many studies analyzing *Taq1A* polymorphism in subjects that are from different ethnical groups and are addicted to different substances, and the meta-analyses show an association of this polymorphism with addiction [6,7,8]. The biological bases for choosing polymorphism are therefore unquestionable. It should be remembered, however, that addiction is a multifactorial disease, and in addition to the genetic component, we must take into account psychological factors, preferably a relation between them. Therefore, the authors of the study took into consideration psychological factors associated with personality traits and anxiety.

In the last two decades, the Five Factor Model, also known as Big Five personality traits, was particularly popular among researchers dealing with personality disorders [7,8,9,10]. This model was created on the basis of psychological studies on the personality structure as a model of the primarily “healthy” personality. Nevertheless, it also assumes that particular configurations of an extremely low or high severity of “correct” traits may be related to personality disorders. This often occurs in addictions. The NEO-FFI questionnaire, also known as “Big Five”, distinguishes the following factors which describe the human personality [11]: openness to experience, conscientiousness, extraversion, agreeableness, and neuroticism. People with high neuroticism show a high tendency towards mood changes, and often experience feelings of anxiety, worrying, anger, fear, frustration, jealousy, guilt, envy, depressive moods and loneliness. [12,13]. As in the case of harm avoidance (HA), neuroticism is linked with the serotonergic system [14,15]. Openness is a personality trait associated with intelligence and divergent thinking. It has been found that openness depends on the function of dopamine, especially in the prefrontal cortex [16]. Conscientiousness is a quality defined as a tendency to control impulses and act in a way which is socially acceptable [17]. Extraversion is characterized by sociability, assertiveness and excitability. Extraverted people may seem more dominant in the social environment, as opposed to people who are locked in this environment [18]. Agreeableness, however, is a tendency towards compassion and cooperation, and also includes attributes such as altruism, trust and other pro-social behaviors.

Another tool that is used in addiction research is the State-Trait Anxiety Inventory (STAI). It is a tool measuring the state of anxiety as well as the trait of anxiety [19].

The aim of this study is to analyze the *Taq1A* polymorphism of the *ANKK1* gene in the group of patients addicted to psychoactive substances and in the control group in consideration of personality traits analyzed by means of the NEO-FFI and STAI questionnaires.

## 2. Materials and Methods 

### 2.1. Subjects

The study was carried out in the Independent Laboratory of Health Promotion, Pomeranian Medical University in Szczecin, after receiving approval from the Bioethics Committee of the Pomeranian Medical University (KB-0012/106/16) and the informed, written consent of the subjects. The study group contained 600 male volunteers, including polysubstance addicted patients (*n* = 299; mean age = 28, SD = 6.45) and healthy controls (*n* = 301; mean age = 22, SD = 4.57). The addicted subjects were recruited at addiction treatment facilities in the province of Lubuskie after at least 3 months of abstaining from drugs. None of the addicted subjects were receiving pharmacotherapy. The control group included healthy, non-addicted subjects. 

Both groups were examined by the psychiatrist using the Mini-International Neuropsychiatric Interview (M.I.N.I). The ICD-10, the State-Trait Anxiety Inventory (STAI) and the NEO Five-Factor Inventory (NEO-FFI) questionnaires were administered. STAI measures anxiety as a state (A-state) including fear, discomfort, and the arousal of the autonomic nervous system occurring temporarily in relation to a particular situation as well as the trait of anxiety (A-Trait), which may be described as a permanent and enduring disposition to experience stress, worries, and discomfort. The Personality Inventory (NEO Five-Factor Inventory, NEO-FFI) includes 6 components for each of the five traits—Neuroticism (Anxiety, Hostility, Depression, Self-consciousness, Impulsiveness, Vulnerability to stress), Extraversion (Warmth, Gregariousness, Assertiveness, Activity, Excitement Seeking, Positive Emotion), Openness to experience (Fantasy, Aesthetics, Feelings, Actions, Ideas, Values), Agreeableness (Trust, Straightforwardness, Altruism, Compliance, Modesty, Tendermindedness), Conscientiousness (Competence, Order, Dutifulness, Achievement striving, Self-discipline, Deliberation) [20]. The results of the tests NEO-FFI and STAI were given as sten scores. The conversion of raw score into the sten scale was performed according to Polish norms for adults, in which it was assumed that: stens: 1–2—very low scores; 3–4—low scores, 5–6—average scores; 7–8—high scores, 9–10—very high scores. DNA was provided from the whole blood aspirated from the elbow vein.

### 2.2. Genotyping

The genomic DNA was isolated from venous blood in compliance with standard procedures. The PCR method was used to genotype samples.

All genotyping was performed with the fluorescence resonance energy transfer using the real-time PCR method on the LightCycler ^®^ 480 II System (Roche Diagnostic, Basel, Switzerland). For polymorphism in the *ANKK1* rs1800497, the following conditions were applied. PCR was performed with 50 ng DNA of each sample in a final volume of 20 µL containing 2 µL reaction mix, 0.5 mM of each primer, 0.2 mM of each hybridization probe, and 2 mM MgCl2 according to the manufacturer’s instructions with initial denaturation (95 °C for 10 min) and then 35 cycles of denaturation (95°C for 10 s), annealing (60 °C for 10 s) and extension (72 °C for 15 s). After amplification, a melting curve was generated by holding the reaction at 40 °C for 20 s and then heating slowly to 95°C. The fluorescence signal was plotted against temperature to provide melting curves for each sample. Peaks were obtained at 58.95 °C for the T (A1) allele and at 67.17 °C for the C (A2) allele.

### 2.3. Statistical Analysis

Analysis of the genotype frequency in regard to its compliance with the Hardy-Weinberg equilibrium (HWE) was performed with the HWE software (http://www.oege.org/software/hwe-mr-calc.html). The rest of the analysis was performed using STATISTICA 13 (Tibco Software Inc., Palo Alto, CA, USA) for Windows (Microsoft Corporation, Redmond, WA, USA).

Differences for particular traits between healthy controls and addicted subject were analyzed.

The distribution of the analyzed variables did not have a normal distribution. To determine whether a subject had anxiety or the severity of this possible anxiety, the NEO Five Factor Inventory (Neuroticism, Extraversion, Openness, Agreeability Conscientiousness) were measured and compared using U Mann-Whitney test.

However, for non-parametric qualities including the ANKK1 *Taq1A* polymorphism (frequency of genotypes and alleles), the chi square test (χ^2^) was applied.

Not all assumptions required for the ANOVA analysis have been met. The assumption about the normal distribution was not fulfilled for all dependent variables but the variance was the same (Levene test *p* > 0.05). Because the number of subjects in groups was also large, it was therefore decided to use multivariate analysis of main effects Multi-factor ANOVA. The test was used to show an association between personality traits (STAI, NEO Five Factor Inventory) and the addiction factor or its absence and the ANKK1 rs1800497 polymorphism (personality traits × control and addicted subjects × genetic feature). 

The Bonferroni multiple comparisons correction was applied for the Mann-Whitney U test in order to exclude the issue of multiple repetitions. The accepted level of significance was 0.0071 (0.05/7).

## 3. Results

The genotypes and alleles frequency distributions were obtained using the Hardy-Weinberg equilibrium (Table 1).

Compared to the controls, the case group subjects had significantly higher scores on the STAI state scale (M = 5.90 vs. M = 4.69, *p* ≤ 0.0071), STAI trait scale (M = 7.11 vs. M = 5.17, *p* ≤ 0.0071), Neuroticism scale (M = 6.73 vs. M = 4.67, *p* ≤ 0.0071), Openness scale (M = 5.01 vs. M = 4.53, *p* ≤ 0.0071), and lower scores on the Extraversion scale (M = 5.76 vs. M = 6.37, *p* ≤ 0.0071), Agreeability scale (M = 4.30 vs. M = 5.59, *p* ≤ 0.0071) and Conscientiousness scale (M = 5.59 vs. M = 6.08, *p* ≥ 0.0071) (Table 2).

No statistically significant difference was found between the addicted subjects and the control group in the frequency for the *ANKK1 Taq1A* genotypes (C/C 0.63 vs. C/C 0.66, C/T 0.33 vs. C/T 0.31, T/T 0.04 vs. T/T 0.02, χ^2^ = 2.19, *p* = 0.335) and the frequency of *ANKK1 Taq1A* alleles (C 0.79 vs. C 0.82, T 0.21 vs. T 0.18, χ^2^ = 1.27, *p* = 0.260) (Table 3).

### ANKK1 Taq1A Variant Interaction STAI Scale

The Multi-factor ANOVA of addicted subjects and control subjects were found for the STAI state scale (F_1,593_ = 40.07, *p* < 0.000, η^2^ = 0.063, observed power = 0.999), STAI trait scale (F_1,592_ = 111.31, *p* < 0.000, η^2^ = 0.158, observed power = 1.00), Neuroticism scale (F_1,594_ = 141.45, *p* < 0.000, η^2^ = 0.192, observed power = 1.00), Extraversion scale (F_1,594_ = 13.28, *p* < 0.000, η^2^ = 0.022, observed power = 0.95), and Agreeability scale (F_1,594_ = 62.54, *p* < 0.000, η^2^ = 0.095, observed power = 1.00). The *ANKK1 Taq1A* main effects approximated to the statistical significance for the STAI trait scale (F_2,592_ = 4.43 *p* = 0.015, η^2^ = 0.015, observed power = 0.76) and the STAI state scale (F_2,593_ = 2.69, *p* = 0.068, η^2^ = 0.009, observed power = 0.53), Neuroticism scale (F_2,594_ = 2.47, *p* = 0.085, η^2^ = 0.008, observed power = 0.50). The means and standard deviations of all the STAI and the NEO-FFI scores in the cases and the controls, and in different ANKK1 Taq1A variants, are presented in Table 4. In addition, the tests of the interactions of addiction status, genetic variant, and personality traits scores are also presented in Table 4.

In the analysis of factors, the relationship between the ANKK1 Taq1A polymorphism and the results on the STAI trait scale approximated to the statistical significance explanation at an approximate level of 1.5%. However, the observed power of interaction effect was 0.76. The phenotype variance found an explanation at a greater level of 16% for the results on the STAI trait scale for traits observed in the analyzed people such as addiction or its absence. In this case, the observed power of interaction was more than 0.99.

Additionally, our power calculation had more than 0.95 observed power to detect addicted subjects and control subjects’ main effects and their interaction effect of the studied STAI state (6% of the phenotype variance found explanation) and NEO Five Factor Inventory Neuroticism scale (19% of the phenotype variance found explanation), Extraversion scale (2% of the phenotype variance found explanation), and Agreeability scale (9% of the phenotype variance found explanation).

## 4. Discussion

The main focus of our study was to combine personality traits measured by the NEO-FFI and STAI inventory, as well as genetic factors in the contextual occurrence of addiction.

We found many important associations concerning the above-mentioned factors. Compared to the controls, the case subjects had significantly higher scores on the scale of the STAI state, STAI trait, Neuroticism, Openness, as well as lower scores on the scales of Extraversion, Agreeability, and Conscientiousness.

The main effects that the addicted subjects and control subjects showed were found for the STAI state, STAI trait, Neuroticism scale, Extraversion scale, and Agreeability scale. The *ANKK1 Taq1A* main effects approximated to statistical significance for the STAI trait, the STAI state, and Neuroticism.

No statistically significant difference was found between the addicted subjects and the control group in the frequency for the *ANKK1 Taq1A* genotypes and alleles.

Multi-factor ANOVA of addicted subjects and control subjects and the *ANKK1 Taq1A* variant interaction approximated to the statistical significance for the STAI state.

Our study shows that the scores in the STAI inventory differ significantly between cases and controls.

Anxiety disorders often co-exist with substance addiction and are more prevalent in families with a problem of psychoactive substances use [21]. Anxiety-impulsive personality traits in people affected by substance use disorders and in their families achieve higher values than in the control group. Anxiety-impulsive personality traits are a possible endophenotype in the risk of developing addiction to cocaine or amphetamine [22]. People with higher levels of anxiety are more susceptible to developing substance addiction. Studies confirm the association between anxiety traits measured with STAI and addiction [23]. Addicted patients had a higher score not only in the STAI inventory, but also on the depression scale and a lower score on the tolerance scale (distress tolerance). Coping with stress and negative mood states is a common motive for the use of psychoactive substances among severe addicts [24].

Research aimed at the analysis of the rate of evolutionary changes in genes, indicate high dynamics of evolutionary changes in the genes of the dopaminergic system, including DRD2 and *ANKK1*. The *ANKK1* gene A1 allele impacts D2 receptor availability in the striatum and is associated with anxiety symptoms lasting from early childhood [25].

The analysis of the results of the NEO-FFI inventory shows significant differences between addicts and controls. The scores regarding neuroticism and openness were higher, and extraversion, compliance and conscientiousness scores were lower in addicts than in the control group. Research shows the leading role of personality traits in the problematic use of substances. People who abuse psychoactive substances, and their relatives who do not suffer from this disease, have higher measures of stress sensitivity than people in the control group, which suggests that neuroticism may be an endophenotype in disorders associated with the use of substances. A study conducted by Terracciano and colleagues [26] shows that low scores of conscientiousness and high scores of neuroticism show an association with the use of numerous psychoactive substances, i.e., tobacco, heroin, and cocaine. People who use cannabis have low scores on the scale of conscientiousness, but they have average results on the scale of neuroticism and high results on the scale of openness, which is a distinguishing feature of the users of this substance. Analysis of tobacco smokers [27] revealed that low scores of neuroticism and openness were associated with tobacco abstinence, and high scores of neuroticism and low scores with regard to agreeableness and conscientiousness were associated with predictors of the worst results, including more cigarettes being smoked on a daily basis.

Following the reports from the literature, it can be said that personality traits may become a predisposing addiction factor. The most widely described is impulsivity, which we define according to Baratt as ‘acting under pressure of the moment”. This is a psychological characteristic describing a given patient, which may become the starting point for defining the endophenotype [28]. This trait may be broken down into choice-making and motoric quality. The one regarding choice-making determines the increased tendency to make ill-judged decisions. Experimenting with various substances and behaviors, which may lead to the development of addictions, is such a decision in the case of addicted patients. A clear combination of this feature with different profiles of addiction may be observed. Drug addicted subjects are characterized by a greater intensity of impulsivity compared to people treated for alcohol dependence [29]. What is commonly described is the so-called “novelty seeking” as a characteristic quality of addicts, showing strong correlations with another dopaminergic system polymorphism i.e., *DRD4*, where the number of tandem repeats is connected by a functional polymorphism with the need for adventure seeking and linked with receptor sensitivity to dopamine uptake [30]. In our study we analyzed openness to new experiences in this context. However, this data cannot be considered separately without analyzing other personality traits measured by means of psychometric scales, which was also applied in our studies.

Dopaminergic transmission was suggested to be associated with novelty seeking [31,32] - a trait associated with dependence [33,34] and relapse [35]. Additionally, extraversion is a personality trait which was linked with the dopaminergic system. It was revealed in the twin studies that each trait is differently influenced by genetics—from 25% to 61% [36].

Dopaminergic conductivity plays an important role in shaping the reward phenomenon in response to drugs. Although a variety of methods were applied and various groups of patients were studied, the study results point to a certain role of *DRD2* gene polymorphisms in the addiction foundation [37].

The role of the dopaminergic system is clearly described in the literature in the context of the reward system in addiction and confirmed by animal studies and neuroimaging. Studies on animals deprived of the D1 dopamine receptor have highlighted the role of these receptors in shaping the reward effect. Mice with no D1 receptors showed a non-disturbed reaction in the cocaine-mediated conditioned place preference testing in which cocaine intake was reduced [38]. However, animals deprived of the D2 receptor showed a suppressed reaction in the reward system in the conditioned place preference testing, both to cocaine and opiates [39]. Some information is provided by imaging using the PET technique. What it enabled to find was that the reduced D2/D3 ratio of dopamine receptors was a marker for impaired reward perception, even during periods of drug abstinence [39].

It is suggested that the role of D1 receptors compared to D2 is different in shaping the reward effect in response to addictive substances [40]. The striatal D1 receptors seem to promote both the response itself in the form of a reward effect and sensitization to psychostimulants, in contrast to D2 receptors which suppress these processes. These observations seem to coincide with the results of previous studies in which an association between reduced availability of the D2 receptor and some addictions, including cocaine addiction, was found [41]. Earlier reports also pointed to reduced availability of this receptor within the striatum in other addictions [42]. In animal studies carried out by Morgan et al. it was noted that after prior reduction in the density of D2 receptors, individuals showed an increased intake of cocaine in self-administration tests [43]. In contrast, the increased expression of these receptors in the areas responsible for shaping the “reward” in rats was significantly reduced by alcohol intake in similar studies carried out by Thanos et al. [44]. In humans, evidence supporting the hypothesis that reduced availability of the D2 dopaminergic receptors is a factor affecting the increased risk of addiction, was provided as a result of family studies using neuroimaging techniques, in which an increased density of the D2 receptors in parents in the control group was found [45].

The ANKK1 *TaqIA* polymorphism shows its influence on the availability of the D2 receptor in the CNS region [46]. Association studies described a connection between ADHD symptomatology with the presence of the *ANKK1 TaqI A1* allele [47], which confirmed previous observations made in 1991 by Comming et al. in patients with ADHD and their relatives, where the frequency of the *TaqI A1* allele occurrence in the researched group was 49%, whereas in the control group it was 27%. The influence of this polymorphism on the risk of pathological gambling was described in the research on the connection of the *TaqI A1* allele with the occurrence of behavioral dependencies [48]. In the metanalysis of genetic studies related to the group of patients with attention deficit hyperactivity disorder, it was indicated that there is a strong connection of this group of disorders with the *ANKK1 TaqI A1* allele [49]. *TaqI* polymorphism shows association with various addictions [50,51,52,53].

More intense phenotypical traits in heterozygotic individuals in relation to homozygotic ones is what we call heterosis [54]. It was observed that heterozygosity of the *ANKK1 TaqIA* polymorphism influences dopaminergic conductivity, which was reflected in the concentration of dopamine metabolites in the cerebrospinal fluid [55]. What was also measured in this research was the presence of attention disorders. A connection between the intensity of attention deficits with the reduced concentration of the homovanillic acid in the cerebrospinal fluid and the *ANKK1 A1/A2* heterozygosity was indicated. Observations from the studies on the connection between the concentration of dopamine metabolites in the cerebrospinal fluid (HVA—homovanillic acid), density of the D2 receptor and intensity of the response to a methylphenidate (psychostimulant drug) confirmed the hypothesis of the influence of this polymorphism on the availability of the D2 receptor [56]. The studies using the PET technique (glucose marked with a fluorine isotope) on the association between the intensity of metabolism in various regions of the human brain revealed a reduced metabolism in the putamen, nucleus ambiguous, parietal-temporal cortex and visual cortex in the *ANKK1 TaqI A1* allele carriers in relation to the ones with the A2 allele [57], which, on the other hand, was related to the reduced density of the D2 dopamine receptors [41]. The study on personality traits in healthy subjects show the effect of *Taq1A* on Neuroticism in both genders. The association between the *Taq1A A2*/*A2*-genotype and higher Novelty Seeking and lower Reward Dependence was shown in men only [58].

The study has some limitations—it was conducted only on males of Caucasian origin. It is necessary to repeat the analysis for a group of women and for persons of other ethnic origins.

## 5. Conclusions

Compared to the controls, the case group subjects had significantly higher scores on the scale of the STAI state, STAI trait, Neuroticism, and Openness, as well as lower scores on the scales of Extraversion, Agreeability, and Conscientiousness. No statistically significant difference was found between the addicted subjects and the control group in the frequency for *ANKK1 TaqIA* genotypes and alleles. Multi-factor ANOVA of addicted subjects and control subjects and the ANKK1 Taq1A variant interaction approximated to the statistical significance for the STAI state. The main effects for addicted subjects and control subjects was found for the STAI state, STAI trait, Neuroticism scale, Extraversion scale, and Agreeability scale. The *ANKK1 TaqIA* main effects approximated to the statistical significance for the STAI trait and the STAI state and Neuroticism. From all the studies shown above, we may draw a conclusion that *ANKK1* variants act in the dopaminergic system, as presented by other researchers [59,60]. Our study shows not only differences in personality traits between addicted and non-addicted subjects, but also the possible impact of *ANKK1* on given traits and addiction itself. It would also be interesting to include epigenetic factors in our analysis to get a full picture of the correlation between factors modulating predisposition to addiction.

## Figures and Tables

**Table 1 ijerph-16-02687-t001:** Hardy-Weinberg equilibrium of the *ANKK1 Taq1A* alleles frequency in a group of addicted subjects and controls.

	Observed (Expected)	Alleles Frequency	χ^2^	*p* Value
Addicted Subjects (*n* = 299)				
C/C	187 (187.06)	p allele freq (C)= 0.79	0.0	>0.05
C/T	99 (98.87)	q allele freq (T)= 0.21		
T/T	13 (13.06)			
Controls (*n* = 301)				
C/C	199 (201.72)	p allele freq (C)= 0.82	1.13	>0.05
C/T	94 (88.56)	q allele freq (T)= 0.18		
T/T	7 (9.72)			

*p* Value—statistical significance χ^2^—test, *n*—number of subjects. Statistically significant differences are marked in bold print.

**Table 2 ijerph-16-02687-t002:** STAI and NEO Five Factor Inventory results in group of addicted subjects and in controls.

STAI/NEO Five Factor Inventory	Addicted Subjects (*n* = 299) M (SD)	Control (*n* = 301) M (SD)	U Mann-Whitney Z	*p* Value
STAI state	5.90 (2.48)	4.69 (2.14)	**6.39**	**0.0000**
STAI trait	7.11 (2.28)	5.17 (2.18)	**9.62**	**0.0000**
Neuroticism/scale	6.73 (2.18)	4.67 (2.01)	**10.78**	**0.0000**
Extraversion/scale	5.76 (2.14)	6.37 (1.98)	**−3.47**	**0.0005**
Openness/scale	5.01 (2.02)	4.53 (1.61)	**2.91**	**0.0036**
Agreeability/scale	4.30 (1.93)	5.59 (2.09)	**−7.52**	**0.0000**
Conscientiousness/scale	5.59 (2.27)	6.08 (2.15)	−2.62	0.0089

Bonferroni correction was used, and the *p* Value was reduced to 0.0071 (*p* = 0.05/7 (number of statistical tests conducted)). M—mean, SD—standard deviation, *U Mann-Whitney*. Statistically significant differences are marked in bold print. The results are given in sten scores.

**Table 3 ijerph-16-02687-t003:** Frequency of genotypes and alleles of the *ANKK1 Taq1A* gene polymorphisms in addicted subjects and controls.

Group	*ANKK1 Taq1A*				
	Genotypes			Alleles	
	C/C *n* (%)	C/T *n* (%)	T/T *n* (%)	C *n* (%)	T *n* (%)
Addicted subjects *n*= 299	187 (0.63)	99 (0.33)	13 (0.04)	473 (0.79)	125 (0.21)
Control *n* = 301	199 (0.66)	94 (0.31)	7 (0.02)	492 (0.82)	108 (0.18)
χ^2^ (df) *p* Value	2.19 (2) 0.335			1.61 (1) 0.204	

*p*-statistical significance χ^2^—test, *n*—number of subjects. Statistically significant differences are marked in bold print.

**Table 4 ijerph-16-02687-t004:** Differences in *ANKK1 Taq1A* and STAI, NEO Five Factor Inventory between healthy control subjects and addicted subjects.

STAI /NEO Five Factor Inventory			ANKK1 Taq1A			Multi-Factor ANOVA (Main Effects) Assessment of the Impact of Main Factors			
	Addicted Subjects (*n* = 299)	Control (*n* = 301)	C/C (*n* = 386)	C/T (*n* = 193)	T/T (*n* = 20)	Factor	F (*p* Value)	η^2^	Observed Power
STAI state M (SD)	5.90 (2.48)	4.69 (2.14)	5.28 (2.38)	5.19 (2.33)	6.60 (1.96)	x intercept	**F_1,593_ = 952.07** **(*p* = 0.000000)**	**0.616**	**1.000**
						x addicted/control	**F_1,593_ = 40.07** **(*p* = 0.000000)**	**0.063**	**0.999**
						x *ANKK1 Taq*1	F_2,593_ = 2.69 (*p* = 0.068473)	0.009	0.534
STAI trait M (SD)	7.11 (2.28)	5.17 (2.18)	6.14 (2.45)	5.95 (2.38)	7.75 (1.94)	x intercept	**F_1,592_ = 1345.42** **(*p* = 0.000000)**	**0.694**	**1.000**
						x addicted/control	**F_1,592_ = 111.31** **(*p* = 0.000000)**	**0.158**	**1.000**
						x *ANKK1 Taq1*	F_2,592_ = 4.43 (*p* = 0.014729)	0.015	0.761
Neuroticism/scale M (SD)	6.73 (2.18)	4.67 (2.01)	5.66 (2.38)	5.63 (2.25)	7.00 (2.10)	x intercept	**F_1,594_ = 1275.89** **(*p* = 0.000000)**	**0.682**	**1.000**
						x addicted/control	**F_1,594_ = 141.45** **(*p* = 0.000000)**	**0.192**	**1.000**
						x *ANKK1 Taq1*	F_2,594_ = 2.47 (*p* = 0.085571)	0.008	0.496
Extraversion/scale M (SD)	5.76 (2.14)	6.37 (1.98)	6.03 (2.11)	6.11 (2.05)	6.30 (1.69)	x intercept	**F_1,594_ = 1399.58** **(*p* = 0.000000)**	**0.702**	**1.000**
						x addicted/control	**F_1,594_ = 13.28** **(*p* = 0.000291)**	**0.022**	**0.953**
						x *ANKK1 Taq1*	F_2,594_ = 0.42 (*p* = 0.655853)	0.001	0.119
Openness/scale M (SD)	5.01 (2.02)	4.53 (1.61)	4.77 (1.85)	4.77 (1.81)	4.90 (1.89)	x intercept	F_1,594_ = 1066.95 (*p* = 0.000000)	0.642	1.000
						x addicted/control	F_1,594_ = 9.81 (*p* = 0.001823)	0.016	0.878
						*x ANKK1 Taq1*	F_2,594_ = 0.01 (*p* = 0.988612)	0.00003	0.052
Agreeability/scale M (SD)	4.30 (1.93)	5.59 (2.09)	4.94 (2.16)	4.97 (2.05)	5.00 (1.75)	x intercept	**F_1,594_ = 970.40** **(*p* = 0.000000)**	**0.620**	**1.000**
						x addicted/control	**F_1,594_ = 62.54** **(*p* = 0.000000)**	**0.095**	**1.000**
						x *ANKK1 Taq1*	F_2,594_ = 0.23 (*p* = 0.791254)	0.0007	0.087
Conscientiousness/scale M (SD)	5.59 (2.27)	6.08 (2.15)	5.92 (2.24)	5.74 (2.16)	5.15 (2.48)	x intercept	F_1,594_ = 1003.77 (*p* = 0.000000)	0.628	1.000
						x addicted / control	F_1,594_ = 6.79 (*p* = 0.009425)	0.011	0.739
						x *ANKK1 Taq1*	F_2,594_ = 1.15 (*p* = 0.316991)	0.004	0.253

Bonferroni correction was used, and the *p* value was reduced to 0.0071 (*p* = 0.05/7 (number of statistical tests conducted)). M—mean, SD—standard deviation. Statistically significant differences are marked in bold print.

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
