# Peer review of "The Ankyrin Repeat and Kinase Domain Containing 1 Gene Polymorphism (ANKK1 Taq1A) and Personality Traits in Addicted Subjects"

_ijerph, 2019, doi:10.3390/ijerph16152687_

Reviewer 1 Report

Re: The Ankyrin Repeat and Kinase Domain Containing 2 1 gene polymorphism (ANKK1 Taq1A) and 3 personality traits in addicted subjects.

The manuscript examines whether psychological factors/traits modify the suspected association of ANKK1 Taq1A polymorphism with addictive behavior. Accounting for the psychological profile when studying genetic and other biological factors in addiction is necessary to gain a deeper understanding of the complex phenomenon of addiction.  As it stands, the study has several problems and needs improvement in its methodology.

Introduction:

1.       The introduction is long and confusing at times. In particular:

a.       The authors failed to clearly explain the background knowledge on which they based their research question. If they needed to assess the effect of the psychological profile on the association of TaqA1 polymorphism with addictive behavior, they need to clearly show that that association is present, or highly plausible, at least. Yet, the intro cited possible influences on dopamine receptor or RDS, but not on addiction. Particularly confusing is the sentence: “Influence of the polymorphism on the availability of the D2 receptor in the CNS was found [6] in neuroimaging. A relationship between the occurrence of the Taq1A polymorphism and the expression of the DRD2 gene was also found as a result of linking with other DRD2 polymorphisms or the functional influence of the ANKK1 gene on dopaminergic conductivity in the CNS region. The latter variant was emphasized in Huang's works in which the influence of the ANKK1 gene on the expression of the DRD2 through transcription factor NFκB was suggested [7]” because it does not give the slightest hint as to the direction of the effects (could it be protective?)

b.       (Pages 2-3, lines 85-100) this paragraph is irrelevant and should be deleted as the study does not examine impulsivity nor adventure-seeking.

c.       Some of the text describing the NEO-FFI and STAI instruments belong to the methods section and should be moved there.

Methods:

1.       It is not clear what the sentence: “The history of drug addiction was recounted using the Polish version of ICD-10, medical history and the authors’ survey” means, since the cases were already selected from addiction treatment facilities. (page 3, lines 122-123)

2.       More description is needed for the items within each scale of the used instruments and how the scores were calculated.

3.       T-test is very sensitive to the shape of the data. Were the compared variables examined for normality?

4.       ANOVA is a bi-variable test, it is not clear how ANOVA was used to perform the multivariable analysis. (is it ANCOVA?)  

5.       It seems to this reviewer that logistic regression with addiction status as the outcome and (psychological indicators x genetic variant) interaction term, as the independent variable, would be more appropriate and easier to interpret. Another suggestion is to use path analysis with psychological indicators as mediator variables (genetic variant 'right arrow' psychological indicator 'right arrow' addiction status)

Results:

1.       Section 3.1 (page 6, line 180) is not clear. What does “The full model ANOVA of addicted subjects…was found…”, and “The main effects of addicted subjects and control subjects were found for the STAI state scale” mean? The pertinent table 4 is also very busy and confusing.

Conclusion:

1.       Without any functional analysis of ANKK1 and DRD2 the conclusion “that ANKK1 variants act in the dopaminergic system, as presented by other researchers, and that ANKK1 gene should be analyzed as a modulator of the DRD2 gene function.” is far-fetched and not supported by the data of this study.

The manuscript needs some editing for language and clarity.

Author Response

ANSWER

Dear Reviewer,

We would like to thank you for your valuable comments on the article. Below you will find our reply to your review. All changes are with a description or a comment and changes have been made to the manuscript (track changes in the tracking group on the review tab).

Introduction:

1.       The introduction is long and confusing at times. In particular:

a.     The authors failed to clearly explain the background knowledge on which they based their research question. If they needed to assess the effect of the psychological profile on the association of TaqA1 polymorphism with addictive behavior, they need to clearly show that that association is present, or highly plausible, at least. Yet, the intro cited possible influences on dopamine receptor or RDS, but not on addiction. Particularly confusing is the sentence: “Influence of the polymorphism on the availability of the D2 receptor in the CNS was found [6] in neuroimaging. A relationship between the occurrence of the Taq1A polymorphism and the expression of the DRD2 gene was also found as a result of linking with other DRD2 polymorphisms or the functional influence of the ANKK1 gene on dopaminergic conductivity in the CNS region. The latter variant was emphasized in Huang's works in which the influence of the ANKK1 gene on the expression of the DRD2 through transcription factor NFκB was suggested [7]” because it does not give the slightest hint as to the direction of the effects (could it be protective?)

Thank you for this valuable comment which was taken into account. This part of  the manuscript was clarified – page 2, lines 52-54.

b.     (Pages 2-3, lines 85-100) this paragraph is irrelevant and should be deleted as the study does not examine impulsivity nor adventure-seeking.

Thank you for this comment. The paragraph was deleted.

c.      Some of the text describing the NEO-FFI and STAI instruments belong to the methods section and should be moved there.

Thank you for this valuable comment which was taken into account. Deatils concerning NEO-FFI and STAI were removed from Introduction (page 2, lines 64-65, 77-78), and added to Methods section (page 3, lines 95-105).

Methods:

1.     It is not clear what the sentence: “The history of drug addiction was recounted using the Polish version of ICD-10, medical history and the authors’ survey” means, since the cases were already selected from addiction treatment facilities. (page 3, lines 122-123)

Thank you for this comment particularly because of the fact that the selection criteria of addicted subjects is a very important issue when a research study is scheduled. We wanted to get as much data as possible from each subject, hence we used many tools to gather information (ICD-10, medical history and the authors’ survey, STAI and NEO-FFI). Addiction is a heterogeneous disease, which is another reason to gather all this data, so we could in the future analyze more homogenous subgroups of subjects.

2.     More description is needed for the items within each scale of the used instruments and how the scores were calculated.

Thank you for the comment which was taken into account. Details concerning NEO-FFI and STAI were added to Methods section (page 3, lines 95-105). We would like to add that all the subject has been examined by psychiatrist, using validated tests.

3.     T-test is very sensitive to the shape of the data. Were the compared variables examined for normality?

Thank you for this comment. During the selection of the test t, it was checked graphically whether the curve is close to the normal distribution. However, after the Kolmogorov-Smirnov test, the condition of the normal distribution of personality traits variable was rejected. Therefore, the Mann-Whitney U test was used and the results were given in the Z scale (page 3, lines 127-130).

4.     ANOVA is a bi-variable test, it is not clear how ANOVA was used to perform the multivariable analysis. (is it ANCOVA?) 

Thank you for this comment. Indeed ANOVA explains the relationship between two variables, the author made a mistake in naming. The Multi-factor ANOVA was used for the analysis. The most common model is the two-factor model (two-factor analysis of variance), although these factors can be much more.  ANOVA was changed to Multi-factor ANOVA (page 1, line 32; page 3, line 136; page 5, line 168; page 9, line 207). Table 4 was rearranged (page 7, line 179).

5.     It seems to this reviewer that logistic regression with addiction status as the outcome and (psychological indicators x genetic variant) interaction term, as the independent variable, would be more appropriate and easier to interpret. Another suggestion is to use path analysis with psychological indicators as mediator variables (genetic variant 'right arrow' psychological indicator 'right arrow' addiction status)

Thank you for this comment. Addiction or lack of it was not used as an explanatory variable, which would allow to conduct the analysis on qualitative variables and allow to change the influence of explanatory factors in this way.

Results:

1.     Section 3.1 (page 6, line 180) is not clear. What does “The full model ANOVA of addicted subjects…was found…”, and “The main effects of addicted subjects and control subjects were found for the STAI state scale” mean? The pertinent table 4 is also very busy and confusing.

Thank you for this valuable comment which was taken into account. The paragraph referring to the overall Multi-factor ANOVA analysis was removed and the impact of the individual factors was left (page 5, lines 167-168). The evaluation data of the overall ANOVA Multi-factor analysis from Table 4 was also removed (page 7, line 179).

Conclusion:

1.     Without any functional analysis of ANKK1 and DRD2 the conclusion “that ANKK1 variants act in the dopaminergic system, as presented by other researchers, and that ANKK1 gene should be analyzed as a modulator of the DRD2 gene function.” is far-fetched and not supported by the data of this study.

Thank you for this valuable comment which was taken into account (page 1, lines 40-42; page 11, lines 328-331).

2.     The manuscript needs some editing for language and clarity.

Thank you for this valuable comment which was taken into account.

Reviewer 2 Report

This paper is an interesting examination of the relationship between a genetic polymorphism and several personality traits, where the comparison was between multiple addicts and a control group. The numbers involved are reasonable for a study of this kind.

The paper is quite well written - perhaps a bit turgid. Sometimes I had queries about what was meant (e.g line number 61), and sometimes it seemed a little over-wordy (eg line numbers 90-92). Nothing that a little editorial /peer-review input could not fix.

However I did have some doubts which are more significant.

The paper starts by saying that ANKK1 is strongly associated with addiction (and the literature supports that). Yet there is no significant difference in allele frequency between the addicted group and the control group. Why not? If it is just that the numbers are too small to show this up, this should be explained. The p values in Table 3 look consistent with that explanation. Most of the more convincing papers about the association are meta-analyses with much higher numbers.

The  p values for differences in personality scores between the groups are hugely significant. Yet data of this kind seldom gives such clear results, and the numbers in the study are good but not huge. I am wondering whether the appropriate statistical methods were used. In particular  line 144 discusses "parametric qualities"   where I think what the authors may mean "continuous variables". Parametric methods such as T-tests and ANOVA should not be used unless the variables being analysed are normally distributed. At the very least the authors need to show that each variable (the scores for the various personality traits) fits the criteria for normal distribution, as part of the validation of these results.

The authors stated that both the control group and the addicted group were drug free (3 months) at the time of testing. Do they mean opiate-free or have they included tobacco and alcohol? Most multiple addicts smoke. Very often clinicians feels that this is the least of their worries, and dislike adding to their stresses by insisting on smoking cessation. Smoking status is important in studies of this kind because smoking lowers monoamine oxidase activity in the brain, affecting signalling pathways. It is possible that there was a big difference in smoking rates between the groups. Similar considerations apply to alcohol. Please clarify. This could be a confounding variable.

Having thought about what Table 4 means, I interpret it as saying that the "addiction vs control" difference accounted for most of the differences in personality scores and the ANKK1 variants accounted for only a small portion of the explanation. If I've got that right - it could be expressed more clearly in the discussion.  If I've got it wrong - maybe other readers will too. What are the p values that apply to whether there is an association between particular genotypes  and personality traits. That might be easier for the general reader...

I was uneasy about the paper's conclusions. For instance if neuroticism is associated with the serotonergic system (line 77) why should an association of ANKK1 and neuroticism then lead to the conclusion that this is mediated through the dopaminergic system. The literature already says that ANKK1 and the functioning of the dopaminergic system are closely linked, as you note - I was left feeling somewhat doubtful that this study really showed  ANKK1's influence on the dopaminergic system or on addiction. The authors will need to explain their thinking more clearly.

Author Response

ANSWER

Dear Reviewer,

We would like to thank you for your valuable comments on the article. Below you will find our reply to your review. All changes are with a description or a comment and changes have been made to the manuscript (track changes in the tracking group on the review tab).

1.     The paper is quite well written - perhaps a bit turgid. Sometimes I had queries about what was meant (e.g line number 61), and sometimes it seemed a little over-wordy (eg line numbers 90-92). Nothing that a little editorial /peer-review input could not fix.

Thank you for this valuable comment which was taken into account, and those parts were removed

However I did have some doubts which are more significant.

2.     The paper starts by saying that ANKK1 is strongly associated with addiction (and the literature supports that). Yet there is no significant difference in allele frequency between the addicted group and the control group. Why not? If it is just that the numbers are too small to show this up, this should be explained. The p values in Table 3 look consistent with that explanation. Most of the more convincing papers about the association are meta-analyses with much higher numbers.

Thank you for this valuable comment. As we know one SNP in not a marker for addiction. Our study group has sufficient power to show this kind of association. This result might be impacted by other factors like for example epigenetic changes, which we will study in the future.

3.     The  p values for differences in personality scores between the groups are hugely significant. Yet data of this kind seldom gives such clear results, and the numbers in the study are good but not huge. I am wondering whether the appropriate statistical methods were used. In particular  line 144 discusses "parametric qualities"   where I think what the authors may mean "continuous variables". Parametric methods such as T-tests and ANOVA should not be used unless the variables being analysed are normally distributed. At the very least the authors need to show that each variable (the scores for the various personality traits) fits the criteria for normal distribution, as part of the validation of these results.

Thank you for this important comment. The distribution of the analyzed variables did not have normal distribution. Thus, variables measuring anxiety and NEO-FFI traits are compared using U Mann-Whitney test. (page 3, lines 127-130).

Not all assumptions required for the ANOVA analysis have been met. The assumption about the normal distribution is not fulfilled for all dependent variables but the variance is the same (Levene test p> 0.05), also the number of subjects in groups is large, therefore it was decided to use multivariate analysis of main effects Multi-factor ANOVA. The test was used to show association between personality traits (STAI, NEO Five Factor Inventory) and the addiction factor or its absence and the ANKK1 rs1800497 polymorphism (personality traits × control and addicted subjects x genetic feature) (page 3, lines 133-139).

Table 4 was rearranged (page 7, line 179).

Commentary for the use of ANOVA in the absence of normal distribution and graphical interpretation of data close to the normal distribution: ANOVA is quite resistant to breaking the normal distribution assumption and can be ignored due to the central limit theorem, according to which the distribution of the mean of the sample tends to a normal distribution, regardless of the distribution of the variable in the population. (Lindman, H. R. 1974. Analysis of variance in complex experimental designs. Oxford, England: W. H. Freeman & Co.)

4.     The authors stated that both the control group and the addicted group were drug free (3 months) at the time of testing. Do they mean opiate-free or have they included tobacco and alcohol? Most multiple addicts smoke. Very often clinicians feels that this is the least of their worries, and dislike adding to their stresses by insisting on smoking cessation. Smoking status is important in studies of this kind because smoking lowers monoamine oxidase activity in the brain, affecting signalling pathways. It is possible that there was a big difference in smoking rates between the groups. Similar considerations apply to alcohol. Please clarify. This could be a confounding variable.

Thank you for this important concern. It was our main thought to recruit only subjects completely free from any psychoactive substances to this project. So, none of analyzed subjects smoked cigarettes (for at least 3 months), or drink alcohol.

5.     Having thought about what Table 4 means, I interpret it as saying that the "addiction vs control" difference accounted for most of the differences in personality scores and the ANKK1 variants accounted for only a small portion of the explanation. If I've got that right - it could be expressed more clearly in the discussion.  If I've got it wrong - maybe other readers will too. What are the p values that apply to whether there is an association between particular genotypes  and personality traits. That might be easier for the general reader...

Thank you for this valuable insight Table 4 was rearanged (page 7, line 179).

6.     I was uneasy about the paper's conclusions. For instance if neuroticism is associated with the serotonergic system (line 77) why should an association of ANKK1 and neuroticism then lead to the conclusion that this is mediated through the dopaminergic system. The literature already says that ANKK1 and the functioning of the dopaminergic system are closely linked, as you note - I was left feeling somewhat doubtful that this study really showed  ANKK1's influence on the dopaminergic system or on addiction. The authors will need to explain their thinking more clearly.

Thank you for this valuable comment which was taken into account (page 1, lines 40-42; page 11, lines 328-331).

Round  2

Reviewer 1 Report

The manuscript is improved somewhat. However, it still needs some clarifications.

Introduction:

1.      The introduction has improved somewhat. However, the connection between ANKK1 Taq1A and addiction is still not clear. References 6, 7, 8 do not seem to belong to the statement “There is many studies…association of this polymorphism with addiction”. (Page 2, lines 52-54)

Methods:

1.      It is not clear what is meant by: “The development and validation of a structured diagnostic psychiatric interview for DSM-IV and ICD-10, the State-Trait Anxiety Inventory (STAI) and the NEO Five-Factor Inventory (NEO-FFI) questionnaires were administered.” (Page 2, lines 93-95)

2.      A brief description of sten scores is preferred, as it is not a familiar scale for the general reader. Also, they should be added to the footnote of Table 2.

3.      It is still not clear what is meant by “The history of drug addiction was recounted using the Polish version of ICD-10, medical history and the authors’ survey”. What is the need for the medical history or the survey if the cases were already selected from addiction treatment facilities? (Page 3, lines 106-107)

4.      Page 3, line 140, change student t-test to Mann-Whitney U

Results and conclusions:

1.      It is not clear how the authors concluded that a three-dimensional relation exists between addiction, ANKK1 Taq1A polymorphism, and personality traits when all the p-values for ANKK1 Taq1A in Table 4 are larger than the preset significance level of .0071. Even if they were significant, the conclusion of an “impact of ANKK1 on given traits and addiction itself” is unwarranted based on statistical significance alone and without adjusting for potential confounders.

The manuscript still needs some editing for language and clarity. For example, change “The means (M) and standard deviations (SD) for all the STAI scales and the NEO Five Factor Inventory scale for the ANKK1 Taq1A variant interaction addicted subjects and control subjects are presented in Table 4” to “The means and standard deviations of all the STAI and the NEO-FFI scores in the cases and the controls, and in different ANKK1 Taq1A variants, are presented in Table 4. In addition, the tests of the interactions of addiction status, genetic variant, and personality traits scores are also presented in Table 4”. (Page 6, lines 176-178)

Author Response

Dear Reviewer,

We would like to thank you for your valuable comments on the article. Below you will find our reply to your review. All changes are with a description or a comment and changes have been made to the manuscript (track changes in the tracking group on the review tab).

Introduction:

1.       The introduction has improved somewhat. However, the connection between ANKK1 Taq1A and addiction is still not clear. References 6, 7, 8 do not seem to belong to the statement “There is many studies…association of this polymorphism with addiction”. (Page 2, lines 52-54)

We would like to thank the reviewer fot this valuable catch – the mistake was made on the editing level. The references are numbered properly now.

Methods:

1.       It is not clear what is meant by: “The development and validation of a structured diagnostic psychiatric interview for DSM-IV and ICD-10, the State-Trait Anxiety Inventory (STAI) and the NEO Five-Factor Inventory (NEO-FFI) questionnaires were administered.” (Page 2, lines 93-95)

Thank you for this comment. The part was rephrased (page 2, lines 93-94).

2.       A brief description of sten scores is preferred, as it is not a familiar scale for the general reader. Also, they should be added to the footnote of Table 2.

Thank you for this valuable comment which was taken into account. This part of  the manuscript was clarified – page 3, lines 104-107; the footnote to table 2 was added as well.

3.       It is still not clear what is meant by “The history of drug addiction was recounted using the Polish version of ICD-10, medical history and the authors’ survey”. What is the need for the medical history or the survey if the cases were already selected from addiction treatment facilities? (Page 3, lines 106-107)

Thank you for this valuable comment. Before the recruitment of patients, they were examined by a psychiatrist, but in fact the patients were recruited at the addiction center and this wording may be misleading for the readers, so we decided to remove this sentence from the mnauscript.

4.       Page 3, line 140, change student t-test to Mann-Whitney U

Thank you for this valuable comment which was taken into account, and the change was made (page 3, line 141)

Results and conclusions:

1.       It is not clear how the authors concluded that a three-dimensional relation exists between addiction, ANKK1 Taq1A polymorphism, and personality traits when all the p-values for ANKK1 Taq1A in Table 4 are larger than the preset significance level of .0071. Even if they were significant, the conclusion of an “impact of ANKK1 on given traits and addiction itself” is unwarranted based on statistical significance alone and without adjusting for potential confounders.

Thank you for this valuable comment which was taken into account, and the changes were made (page 1, lines 36-42; page 5-6, lines 175-177; table 4; page 9, lines 186, 206, 210-212; page 11, lines 325-334)

2.       The manuscript still needs some editing for language and clarity. For example, change “The means (M) and standard deviations (SD) for all the STAI scales and the NEO Five Factor Inventory scale for the ANKK1 Taq1A variant interaction addicted subjects and control subjects are presented in Table 4” to “The means and standard deviations of all the STAI and the NEO-FFI scores in the cases and the controls, and in different ANKK1 Taq1A variants, are presented in Table 4. In addition, the tests of the interactions of addiction status, genetic variant, and personality traits scores are also presented in Table 4”. (Page 6, lines 176-178)

Thank you for this valuable comment which was taken into account, and the change was made (page 6, lines 177-180)

Reviewer 2 Report

I thank the authors for their efforts to make their thought processes clearer, which has really helped the paper. 

It could still do with a little editing for style, but I will leave that for the journal. 

Author Response

Dear Reviewer,

We would like to thank you for your valuable input to the quality of the article.

This manuscript is a resubmission of an earlier submission. The following is a list of the peer review reports and author responses from that submission.